# Antibacterial and Anti-Inflammatory Properties of a Novel Antimicrobial Peptide Derived from LL-37

**DOI:** 10.3390/antibiotics11060754

**Published:** 2022-06-01

**Authors:** Haiwei Zhuo, Xi Zhang, Maogen Li, Qian Zhang, Yonglan Wang

**Affiliations:** School and Hospital of Stomatology, Tianjin Medical University, Tianjin 300070, China; zhuohaiwei@126.com (H.Z.); limaogendental@163.com (M.L.); zhangqian1218@tmu.edu.cn (Q.Z.)

**Keywords:** antimicrobial peptides, oral biofilms, MC3T3-E1 cells, RAW264.7 cells, anti-inflammatory activity

## Abstract

Peri-implantitis is a pathological condition involving tissues around dental implants that are characterized by inflammation of the peri-implant mucosa and progressive loss of supporting bone. We found that the antimicrobial peptide KR-12-3 (KRIVKWIKKFLR) derived from LL-37 had antibacterial properties against *Streptococcus gordonii*. The purpose of this study was to evaluate its antibacterial and anti-inflammatory activities and its underlying mechanisms. We evaluated the antibacterial activities of antimicrobial peptides in planktonic and biofilm states by measuring their minimum inhibitory concentration, minimum bactericidal concentration, and biofilm susceptibility. The effects of antimicrobial peptides on the production of IL-6 and IL-8 in LPS-stimulated RAW264.7 cells were detected by enzyme-linked immunosorbent assay and other experiments, and their toxicity to MC3T3-E1 cells was also studied. While maintaining low cytotoxicity, KR-12-3 exhibited growth inhibitory effects on *S. gordonii* in planktonic and biofilm states. Lower concentrations of KR-12-3 treatment reduced the production of inflammatory cytokines in LPS-stimulated RAW264.8 cells. The mechanisms underlying the inhibition of biofilm formation and anti-inflammatory effects have been associated with the low expression of related genes. KR-12-3 may be used to develop an antibacterial, anti-infective, and anti-inflammatory drugs for peri-implantitis.

## 1. Introduction

Replacing missing teeth with dental implants as a treatment option has become a popular trend [1]. With the development of dental implants, peri-implantitis continues to appear as a major complication, which has now become an important problem for patients and dentists [2]. There is ample evidence that inflammation of the peri-implant tissue can cause lesions similar to periodontal disease. This inflammatory state not only affects the soft and hard tissues around the implant, but also may eventually cause the implant to loosen or even fall off. Different treatments, mainly surgical and non-surgical, can be used, depending on the specifics of the peri-implant disease [3,4]. Treatments are also inseparable from the application of antibiotics.

There is ample evidence that plaque accumulation is a major cause of peri-implantitis [5,6,7]. Studies have shown that 71.7% of patients with peri-implantitis have pathogens that are resistant to at least one antibiotic [8]. Apart from self-adherence, early-colonizing bacteria also provide settlement surfaces for late ones and contribute to coaggregation among different bacteria [9,10,11,12], and *S. gordonii* is one of the representatives of the early colonizers.

Synthetic cationic peptides are emerging treatments for bacterial infections around implants that are aimed at preventing the development of drug resistance. However, synthetic peptides often have problems such as high synthesis costs and instability [13]. To address this shortcoming, researchers often choose to shorten their sequences [14].

In recent years, LL-37 has attracted extensive attention because of its excellent antibacterial and anti-inflammatory properties [15,16,17,18]. Among its known fragments, KR-12 (KRIVQRIKDFLR) has potential as a template or sequence unit for the design of peptides because of its shorter sequence duration and low cytotoxicity [19]. In this study, KR-12-3 (KRIVKWIKKFLR) was obtained using KR-12 as a template and substituting 22Q23R26D for 22K23W26K. 

More patients and dentists are now affected by peri-implantitis. Prompt detection and appropriate treatment can reduce the occurrence or slow down the progression of the disease. The difficulty of treating this disease is self-evident. It is generally accepted that the risk of developing peri-implantitis increases with time after implant placement [20]. It would be better if there were new strategies to reduce the risk of peri-implantitis, especially in the early stages of implant placement. As the culprit of peri-implantitis, the plaque biofilm on the implant surface has become the main target of our research. That is to say, our research aims to find effective antibacterial agents that can inhibit or kill the early colonized bacteria on the implant surface. It would be better if it has a certain anti-inflammatory activity.

Experiments were conducted to study the antibacterial activity of KR-12-3 against *S. gordonii*. We also investigated the structural characterization of KR-12-3 and its effects on oral biofilm formation at an early stage. MC3T3-E1 cells and LPS-stimulated RAW264.8 cells were used to figure out the cytotoxicity and anti-inflammatory activity of KR-12-3. The purpose of this study was to evaluate the ability of KR-12-3 to interfere with biofilm formation, to test its biocompatibility and anti-inflammatory activity, and to determine whether it can be used to prevent peri-implant inflammation. Notably, we found that most researchers focused on the antibacterial capabilities of synthetic peptides and less on anti-inflammatory properties. Therefore, the novelty of this study was to simultaneously explore the anti-inflammatory and antibacterial abilities of peptides. 

## 2. Materials and Methods

### 2.1. Peptide Synthesis

Peptide KR-12-3 emerged after grafting gallic acid to the N-terminus of peptide KR12 (KRIVQRIKDFLR). KR-12-3 was synthesized by standard solid phase peptide synthesis using standard fluoro-benzyloxycarbonyl synthesis by Dechi Biosciences (Shanghai, China). The final chimeric peptide (over 95% purity) was purified by high performance liquid chromatography HPLC and characterized by electron spray ionization. The peptide was dissolved in phosphate buffered saline (PBS) to a specific concentration for subsequent studies. Different methods were used to forecast the basic properties of KR-12-3.

### 2.2. Minimum Inhibitory Concentration (MIC) and Minimum Bactericidal Concentration (MBC)

*S. gordonii* (ATCC No 10558) was obtained from the American Type Culture Collection (ATCC) and cultured aerobically on brain–heart infusion (BHI) agar plates supplemented with 1% yeast extract at 37 °C [21]. The antibacterial properties of KR-12-3 (serial two-fold dilutions from 2500 to 10 μg/mL) against *S. gordoniis* can be partially demonstrated by MIC and MBC [22,23]. PBS was used as the negative control. Different concentrations of KR-12-3 solutions were mixed with bacteria cultured in BHI broth (106 CFU/mL) at a ratio of 1:1, and the total volume of the final mixture was 200 μL. Optical density at 600 nm was measured by a microplate reader to determine bacterial growth after 24 h of anaerobic incubation of the mixture. The MIC value is defined as the absence of visible bacterial growth in the well. Transfer 10 μL of liquid from the wells used for MIC determination to blood agar plates and incubate at 37 °C for 48 h. The final concentration corresponding to no bacterial growth is called MBC [24]. 

### 2.3. Biofilm Susceptibility Assay

The effect of KR-12-3 on biofilm formation was assessed using crystal violet staining. KR-12-3 was co-cultured with *S. gordonii* (5 × 10^7^ CFU/mL) for 24 h in a 96-well plate. Biofilms were fixed with methanol (95%) and stained with 0.5% (w/v) crystal violet, then the stained biofilms were dissolved in ethanol and their optical density at 600 nm was finally measured.

### 2.4. Confocal Laser Scanning Microscopy (CLSM)

CLSM (LSM900, Carl Zeiss, Jena, Germany) was used to investigate whether and how KR-12-3 effected mature biofilms. Bacteria were grown for 24 h in glass-bottom Petri dishes containing liquid medium. KR-12-3(156.25 µg/mL) was added to 1-day-old biofilms. Bacteria grown in liquid medium only were selected as positive controls. Biofilms were rinsed repeatedly with PBS after 24 h of incubation and stained with acridine orange/ethidium bromide (AO/EB) solution. After staining, the images were observed using a confocal laser scanning microscope. The areas were scanned using a 20× objective lens with signals recorded in the green and red channels. 

### 2.5. SEM

Scanning electron microscopy (SEM, Gemini300, Carl Zeiss, Jena, Germany) was performed to analyze the morphology of *S. gordonii* in the presence of KR-12-3. KR-12-3 at the same concentration as previously measured MIC was co-cultured with bacteria (10^8^ CFU/mL) in liquid medium. After 24 h, the co-culture mixture was centrifuged to obtain bacterial precipitation. Then, 1 mL of 2.5% glutaraldehyde was added to fix it for 2 h at room temperature. Finally, bacterial precipitation was dehydrated by gradient (30%, 50%, 70%, 80%, 90%, 95%, and 100%, each concentration 15 min), freeze-dried, and sprayed with gold. Then, the cell morphology was observed by SEM. All experiments were repeated three times.

### 2.6. Cytoxicity

MC3T3-E1 cells obtained from Procell Natural Science & Technology (Wuhan, China) was used as a model of toxicity. Cells were cultured in Dulbecco’s modified eagle medium (DMEM, Gibco) containing 10% (v/v) fetal bovine serum and 3% (v/v) penicillin–streptomycin solution in a constant temperature incubator (37 °C, 5% CO_2_ atmosphere). The medium was replaced every two days. The cells were treated with KR-12-3 or left untreated. The concentration of KR-12-3 co-cultured with the cells was 312.5 μg/mL. The 3-(4,5-dimethylthiazole-2-yl)-2,5-diphenyltetrazoli-umbromide (CCK-8) assay was utilized to measure the cytotoxicity of KR-12-3. To determine the viability of these cells, the absorbance was measured at 450 nm.

### 2.7. Quantitative Reverse Transcription-Polymerase Chain Reaction (RT-qPCR)

Trizol reagent was used to extract total RNA from *S. gordonii* cells. Primers for 16S rRNA, *sspA*, *sspB*, IL-6, IL-8 (Table 1) were designed using Oligo Explorer software and synthesized by Sangon Biotech (Shanghai, China). The first strand of polymer was synthesized by retro-transcription of 2 µL of RNA (Beijing Solarbio Science & Technology Co., Ltd., Beijing, China). Amplify complementary DNA to a final volume of 20 µL in a DNA thermal cycler for 45 cycles. Pre-denaturation was performed at 95 °C for 15 s, and reactions were performed at 95 °C for 10 s, 60 °C for 20 s, and 72 °C for 30 s. The primer sequences of the differentiation markers are shown in Table 1. After the amplification was completed, GAPDH was used as the internal reference gene, and the cycle threshold (CT) value of the corresponding target gene of each sample was obtained by taking the control sample as the standard. When analyzing data for relative quantification, use the formula RQ = 2^−ΔΔCT^ to obtain relative quantification (RQ) values for all target genes. Similarly, RT-qPCR was used to detect the expressions of IL-6- and IL-8-related genes in the cells. The method was the same as that mentioned above.

### 2.8. Cytokine and Inflammation Assays

The cells were treated with KR-12-3 for 24 h after being stimulated with or without LPS (100 ng/mL) [25]. RAW264.7 cells were centrifuged at 1000× *g* for 5 min at 4 °C to obtain the cell supernatant. ELISA was used to identify IL-6 and IL-8 concentrations in the supernatant of RAW264 cells [26].

### 2.9. Statistical Analysis

All experiments used in this study were triplicated and repeated three times on different days. Data were converted to mean ± standard deviation (SD). Statistical analysis of significant differences between groups was performed using ANOVA and independent sample *t*-test (SPSS 22.0), where a value of *p* < 0.05 indicated statistical significance.

## 3. Results

### 3.1. Structural Characterization of Peptides

The basic structural features of the two peptides were shown in Table 2. The peptides possessed hydrophobic residue contents ranging from 41.67 percent for KR-12 to 50% for KR-12-3, net charges ranging from +4 for KR-12 to +6 for KR-12-3, and relative amphipathicities ranging from 0.782 for KR-12 to 0.893 for KR-12-3. The helical wheel projection was performed using the online program HeliQuest: http://heliquest.ipmc.cnrs.fr (accessed on 17 March 2022). Hydrophobic and hydrophilic amino acids are evenly distributed in the peptide chains. The helical wheel diagram for both peptides demonstrated a distinct split between the hydrophilic and hydrophobic areas (Figure 1).

### 3.2. Antimicrobial Activity against S. Gordonii

The MIC and MBC of the two peptides against *S. gordonii* were determined (Table 3). KR-12-3 had stronger inhibitory and bactericidal activities with MIC of 156.25 µg/mL and MBCs of 312.5 µg/mL. The MIC and MBC of KR-12 were 16–32 times higher than those of KR-12-3. KR-12-3 showed more effective antimicrobial activity than KR-12, and it was selected for further evaluations including biofilm assay, cell morphology observation, and cytotoxicity assay.

### 3.3. Biofilm Inhibition

According to the biofilm susceptibility test, KR-12-3 demonstrated high activity against *S. gordonii* biofilm formation. The bacterial biofilms treated with KR-12-3 showed a reduced OD (Figure 2a). As shown in Figure 2b,c, in CLSM, green represents live bacteria, red represents dead bacteria, and yellow represents the proximity of live and dead bacteria. The visual field in the control group was mainly green. In the KR-12-3 group, the green area decreased and the red area increased.

### 3.4. SEM

Surface morphology and intracellular changes in peptide-treated bacterial cells were investigated using SEM. Compared with the smooth surface of the control group, obvious cell wall deformation, corrugation, and damage were observed on the *S. gordonii* surface after 4 h of treatment with the peptide, and bacterial cell wall lesions and intracellular discharge were also observed after treatment (Figure 3).

### 3.5. Effects of the KR-12-3 Peptide on S. gordonii Surface Adhesion Protein sspA/sspB Gene Expression

The results of RT-qPCR showed that the expression of bacterial *sspA* and *sspB* in the KR-12-3 treatment group was significantly different from that in the control group (Figure 4). Studies have shown that *sspA* and *sspB* play important roles in bacterial adhesion and coaggregation. This result may indicate that KR-12-3 inhibits biofilm formation by *S. gordonii* by downregulating the expressions of *sspA* and *sspB* genes.

### 3.6. Cytotoxicity Assay

KR-12-3 and KR-12 at 312.5 µg/mL slightly inhibited MC3t3-E1 cell proliferation relative to the control group, but no significant differences were observed after longer treatment durations (Figure 5a). This was also confirmed by CLSM (Figure 5b). CLSM observation after co-culture of KR-12-3 and Mc3t3-e1 cells showed that the number of cells in the experimental group was less than that in the control group during the first 3 days, but there was no significant difference on the fourth day.

### 3.7. Effect of KR-12-3 and KR-12 on IL-6 and IL-8 Production/Gene Expression in LPS-Stimulated RAW264.7 Cells

RAW264.7 cells were treated with LPS (100 ng/mL) in the absence or presence of peptides. The secretion levels of IL-6 and IL-8 were determined by ELISA. Figure 6a,c shows that LPS significantly (*p* < 0.01) induced IL-6 and IL-8 production in RAW cells. While 20 µg/mL of KR-12-3 significantly (*p* < 0.01) blocked LPS-induced IL-6 and IL-8 production, resulting in approximately 60% and 20% inhibitions of IL-6 and IL-8, respectively. The expression of IL-6 was significantly decreased after KR-12-3/KR-12 treatment at the same concentration. KR-12 reduced the gene expressions of IL-6 and IL-8 but caused only a significant reduction in IL-6 production, and there was no statistical difference in the effect on IL-8 production (Figure 6b,d). Subsequent experiments showed that both peptides induced the downregulation of the IL-6/IL-8-related gene expression.

## 4. Discussion

Typical clinical features of peri-implantitis include hemorrhage/suppuration on probing, increased probing depth, and radiographic signs of bone loss [27]. The weighted mean prevalence of peri-implant mucositis and peri-implantitis were 43% and 22%, respectively, according to a recent meta-analysis [28]. The methods currently used to prevent peri-implantitis mainly consist of mechanical plaque control by the patient (tooth brushing) or professional plaque control such as oral hygiene instruction and mechanical debridement, as well as adjunctive measures (topical and systemic antimicrobial application, spraying sand [29]. However, there is still a lack of sufficient evidence that adjunctive measures can improve the efficacy of professionally managed plaque removal in reducing clinical symptoms of inflammation [30].

Research on the drug treatment of peri-implantitis has varied in recent years [31]. In order to improve the effectiveness of non-surgical treatment of peri-implantitis, antibiotics will never be absent. Nevertheless, systemic antibiotics have many adverse reactions and side effects, such as dysbacteriosis, drug resistance, gastrointestinal reactions, systemic allergic reactions, etc. Therefore, improving post-implant osseointegration and preventing peri-implantitis have attracted much attention. Especially in recent years, the methods of implant surface treatment emerge in an endless stream. Some research has focused on the surface modification of implants or abutments and healing cups/screws. To improve the antibacterial and anti-inflammatory characteristics of titanium (Ti) surfaces, several approaches have been utilized to coat different materials [32,33,34]. At the same time, Sorsa et al. have focused on host modulation therapy (HMT) [35]. Whether this treatment is beneficial for the treatment of peri-implantitis; more clinical studies are urgently needed to confirm.

KR-12-3 is a 12-residue peptide derived from LL-37, a well-known AMP. Structural characterization showed that KR-12-3 has a clear separation of hydrophilic and hydrophobic regions. KR-12-3 contains six polar and six hydrophobic residues. Its net charge is +6. Peptide–cell wall interactions appear to be influenced by attractive forces between lipids on the cell wall surface and charges of opposite nature on the polypeptides. KR-12-3 contains four lysine residues. The electrostatic attraction that exists between these positively charged lysine residues and negatively charged phospholipids may facilitate peptide-cell wall contact. Despite the presence of lipids, the structure of the peptides is unrelated to their biological properties. Our study found that KR-12-3 can disrupt bacterial cell wall, possibly by altering their structure. The antibacterial activity of KR-12-3 against *S. gordoniis* was excellent, which can be proved by the results of antibacterial experiments. SEM results showed that KR-12-3 could damage bacterial cell wall, causing cell wall perforation and bacterial shrinkage. Its ability to inhibit biofilm formation may be achieved through the downregulation of genes associated with bacterial adhesion. In the present study, the effects of KR-12-3 and KR-12 on IL-6/IL-8 production in LPS-stimulated RAW264.7 cells were evaluated and compared. According to the results of ELISA, KR-12-3 reduced IL-6 and IL-8 production in LPS-stimulated RAW264.7 cells at relatively low doses. To determine its underlying mechanism, RT-qPCR was performed. These results indicate that KR-12-3 leads to the downregulation of IL-6/IL-8-related genes. These results are consistent with the results of several studies on LL-37 and its derived peptides. Finally, it seems that KR-12-3 has low cytotoxicity with good antibacterial and anti-inflammatory properties. 

Several studies suggested that LL-37 plays an important role in combating bacterial infection and endotoxin-induced inflammation [36,37,38]. In this study, we found that low concentrations of KR-12-3 exhibited some anti-inflammatory activity. LPS-stimulated macrophages release various pro-inflammatory mediators and cytokines, whose inordinate release may cause pathological changes and tissue damage [39,40,41]. In chronic inflammation, macrophages play an important role in tissue destruction, often producing excessive pro-inflammatory mediators, cytokine production, such as NO, PGE2, TNF-α, IL-1β, etc. [42,43,44] The synergy among these inflammatory mediators accelerates the progression of inflammation. Our results showed that KR-12-3 at low cytotoxic concentrations reduced IL-6 and IL-8 production in LPS-stimulated RAW264.7 cells.

IL-6 signaling pathway activation mode is as follows: First, the IL-6-IL-6R-gp130 complex is formed. Second, Janus kinase (JAK) is recruited and mediates the phosphorylation of signal transducer and activator of transcription 3 (STAT3). Then, the formation of phosphorylated STAT3 homodimers occurs. Activation of the JAK-STAT3 signaling pathway upregulates the expression of IL-6-responsive genes including suppressor of cytokine signaling 1 (SOCS1) and SOCS3 [45]. In addition, JAK phosphorylates the cytoplasmic domain of gp130 at tyrosine 759 (Y759), which acts as the binding site of SH2 domain tyrosine phosphatase 2 (SHP2), further activating the mitogen-activated protein kinase (MAPK) pathway [46]. SOCS1 and SOCS3 directly bind to activated JAK and JAK phosphorylated at Y759, which acts as a negative feedback loop against the JAK-STAT3 and MAPK pathways, respectively [47].

IL8 is induced by pro-inflammatory cytokines, such as TNF-α and IL-1β. Studies have shown that expression of IL-8 is primarily regulated by activator protein-1(AP-1) and/or nuclear factor-κB–mediated transcriptional activity [48]. 

The anti-inflammatory properties of KR-12-3 may be related to JAK-STAT3, MAPK, AP-1, JNK, NF κB, and some other signaling pathways. We also hypothesized that KR-12-3 might also reduce the production of cytokines such as IL-1β and TNF-α.

The strength of this study is to combine the research on the antibacterial and anti-inflammatory properties of peptides, trying to inherit the advantages of natural antibacterial peptides to synthetic peptides. It also provides new ideas for the subsequent transformation and synthesis of peptides. There are some limitations in this study, such as the inability to deeply explore the antibacterial and anti-inflammatory mechanisms of KR-12-3, and the lack of experiments related to peptide stability and pharmacokinetics.

## 5. Conclusions

In summary, the antimicrobial and bactericidal assays of KR-12-3 to *S. gordonii* were studied. KR-12-3 inhibited the growth of *S. gordonii* in planktonic and biofilm states. Lower concentrations of KR-12-3 reduced LPS-stimulated RAW264.8 macrophage inflammatory cytokine production and downregulated the expression of related genes. No apparent cytotoxicity was observed in MC3T3-E1 cells. We, therefore, concluded that KR-12-3 was not significantly cytotoxic, had inhibitory effects on the growth and biofilm formation of *S. gordonii*, and exhibited immunomodulatory properties. However, the activity of KR-12-3 in the oral environment should be fully investigated to assess its potential as an active ingredient in oral-care products.

## Figures and Tables

**Figure 1 antibiotics-11-00754-f001:**
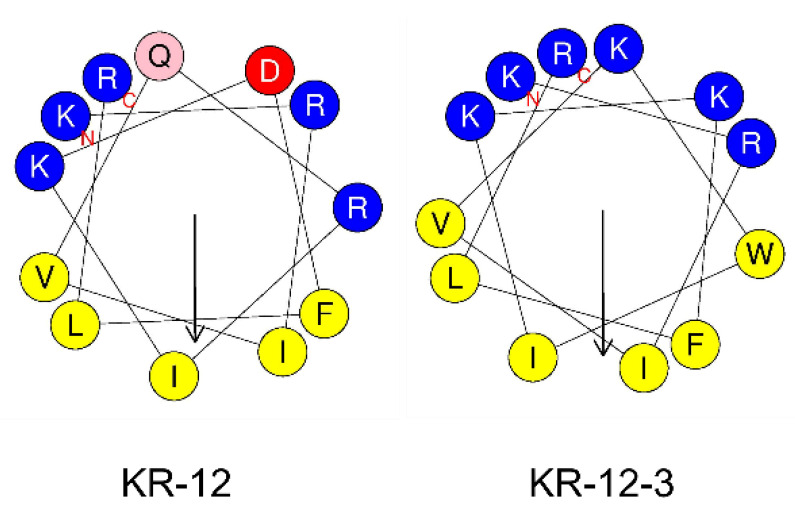
Sequence and helical wheel diagrams of KR-12 and KR-12-3. The down arrows represent the μH vector. Blue circles refer to hydrophilic residues, pink circle refers to glutamine, red circle refers to aspartic acid, and yellow circles refer to hydrophobic residues.

**Figure 2 antibiotics-11-00754-f002:**
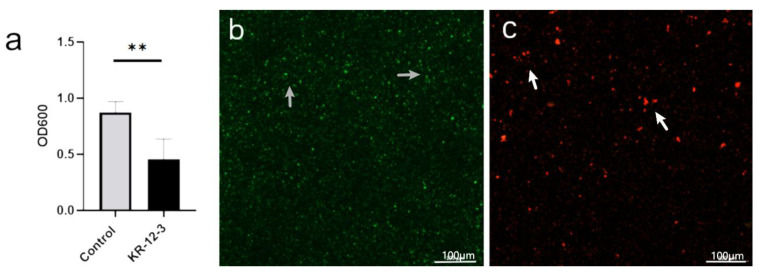
Antimicrobial effects of KR-12-3 against *S. gordonii. S. gordonii* biofilms were treated with KR-12-3 for 24 h (**a**). The data are presented as mean ± SD; *n* = 3. ** *p* < 0.01 compared with control groups. The biofilms were stained and imaged by CLSM. Images of the control group show mature *S. gordonii* biofilms as indicated by grey arrows (**b**). The CLSM image of the KR-12-3 treatment group showed no obvious viable bacteria, only some dead bacteria, as indicated by the white arrows (**c**). The green color represented live bacterial cells, and the red/orange color represented dead bacterial cells. PBS was tested as a buffer control. The experiments were all repeated at least 3 times.

**Figure 3 antibiotics-11-00754-f003:**
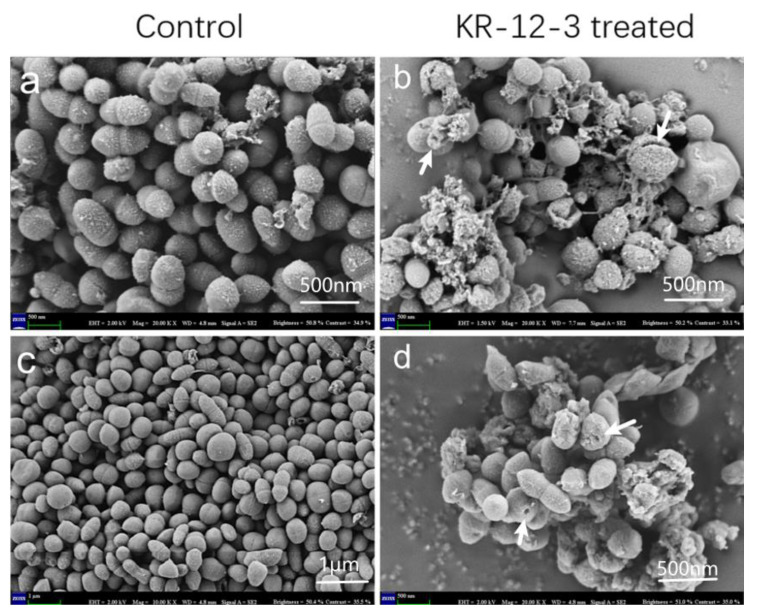
The control group, which lacks peptide treatment, was cultured for 4 h, as shown in (**a**,**c**). (**b**,**d**) were SEM micrographs of *S. gordonii* cells treated with peptide at 1× MICs for 4 h. As indicated by the white arrows, significant cell wall distortion, corrugation, and damage were observed on the surface of *S. gordonii* cells after treatment with the peptide compared with the smooth surface of the control. The experiments were all repeated at least 3 times.

**Figure 4 antibiotics-11-00754-f004:**
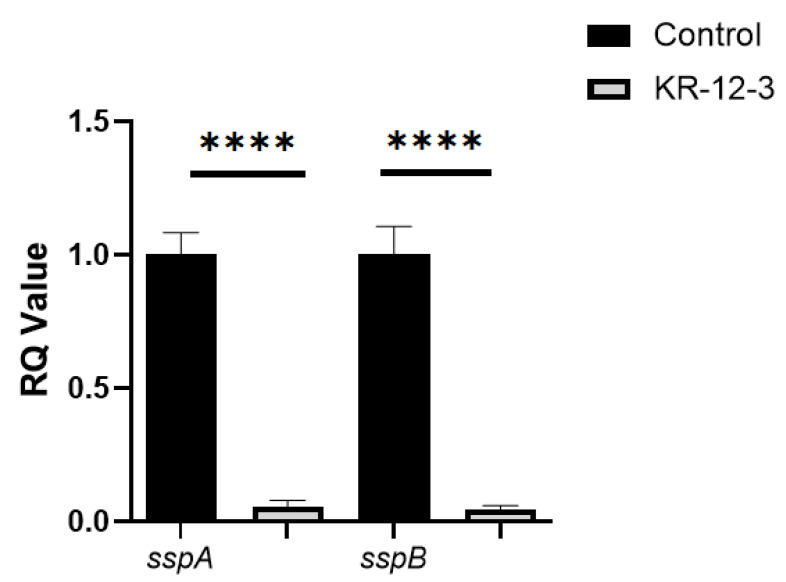
KR-12-3 significantly reduced the expression of *S. gordonii* adhesion genes (*sspA* and *sspB*). **** *p* < 0.0001 compared with control groups. The experiments were all repeated at least 3 times.

**Figure 5 antibiotics-11-00754-f005:**
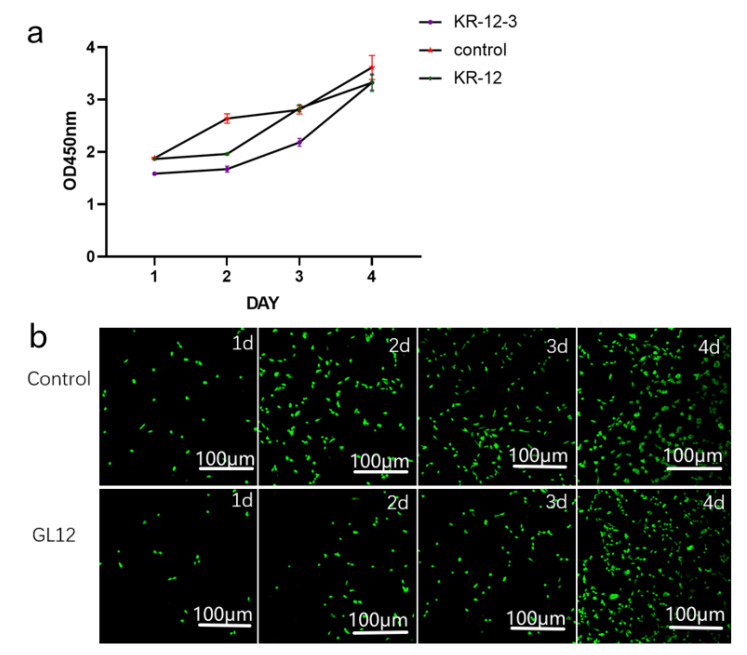
Proliferation curve of MC3T3-E1 cells treated with or without KR-12-3 and KR-12 for 1–4 days (**a**). Data are shown as mean ± SD; *n* = 3. MC3T3-E1 cells stained with AO/EB (green: live; red: dead) after 1–4 days of incubation in the absence (control) and presence of KR-12-3 at 25 µg mL^−1^ (**b**). (**b**) shows the results of 0–4 d of co-culturing of PBS or KR-12-3 and Mc3t3-e1 cells, respectively. The experiments were all repeated at least 3 times.

**Figure 6 antibiotics-11-00754-f006:**
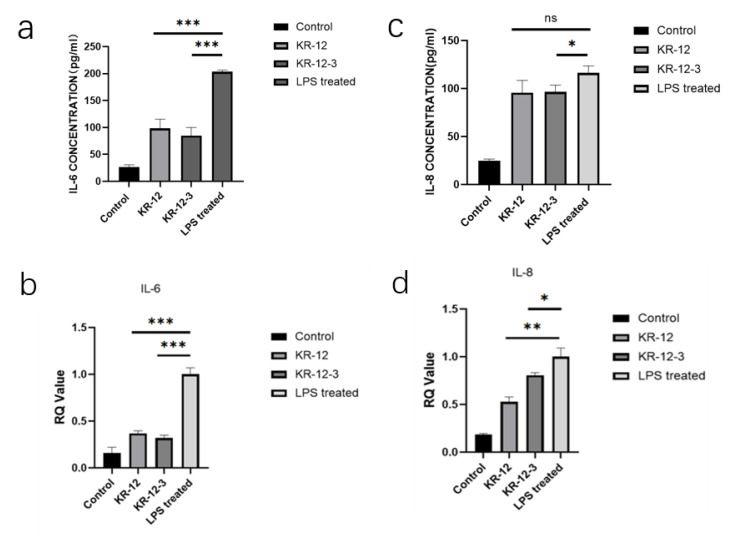
(**a**,**c**) Effects of peptides on LPS-induced production of IL-6 and IL-8 in RAW264. Cells were stimulated with LPS (100 ng/mL) for 24 h in the presence or absence of the peptide (20 μg/mL). After culture, the concentrations of IL-6 and IL-8 in the culture supernatant were determined by enzyme immunoassay. The data are expressed as mean ± standard deviation of independent experiments, * *p* < 0.05 Compared with unstimulated macrophages; ** *p* < 0.01 compared with LPS-stimulated macrophages; *** *p* < 0.001 compared with LPS-stimulated macrophages. (**b**,**d**) Effects of peptides on IL-6 and IL-8 expression in LPS-stimulated RAW264 cells. Cells were stimulated with LPS (100 ng/mL) for 24 h in the presence or absence of peptides. The total mRNA was isolated, and the mRNA expressions of IL-6 and IL-8 were detected by RT-qPCR. Data were expressed as mean ± standard deviation of three independent experiments. ** *p* < 0.001 compared with unstimulated macrophages; * *p* < 0.05 compared with LPS-stimulated macrophages. The experiments were all repeated at least 3 times.

**Table 1 antibiotics-11-00754-t001:** Primer sequences of the differentiation markers.

Gene	Primer Sequence
*sspA*-F	5′-TCCTGACAAACCTGAGACACC-3′
*sspA*-R	5′-TTTAACTTTCAGAGCTTAGTTGCTTTC-3′
*sspB*-F	5′-TCCTGACAAACCTGAGACACC-3′
*sspB*-R	5′-CATCAAAGATGAAACAAGTCTAAGC-3′
16S rRNA-F	5′-AAGCAACGCGAAGAACCTTA-3′
16S rRNA-R	5′-GTCTCGCTAGAGTGCCCAAC-3′
GAPDH-F	5′-GTTGTCTCCTGCGACTTCA-3′
GAPDH-R	5′-GCCCCTCCTGTTATTATGG-3′
IL-6-F	5′-CTGCAAGAGACTTCCATCCAG-3′
IL-6-R	5′-AGTGGTATAGACAGGTCTGTTGG-3′
IL-8-F	5′-GACTTCCAAGCTGGCTGTTG-3′
IL-8-R	5′-GGGTGGAAAGGTGTGGAATG-3′

**Table 2 antibiotics-11-00754-t002:** Structural parameters of KR-12 and KR-12-3. The hydrophobic residue composition (% hydrophobic residues), net positive charge, and relative amphipaticity (μH relative) for each peptide were shown.

Peptide	%Hydrophobic Residues	Net Positive Charge	μH (Relative)	PI
KR-12	41.67	4	0.782	12.20
KR-12-3	50	6	0.893	11.87

**Table 3 antibiotics-11-00754-t003:** In vitro susceptibility of *S. gordonii* to KR-12-3 and KR-12.

Peptide	MIC (µg/mL)	MBC (µg/mL)
KR-12-3	156.25	312.5
KR-12	>2500	>2500

## Data Availability

All data included in this study are available upon request through contact with the corresponding author.

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
