# Peer review of "Antibacterial and Anti-Inflammatory Properties of a Novel Antimicrobial Peptide Derived from LL-37"

_antibiotics, 2022, doi:10.3390/antibiotics11060754_

Round 1
Reviewer 1 Report
The manuscript is well focused on the objectives. It is very interesting, and it improve knowledge about the possible use and application of peptides with antibacterial and anti-inflammatory properties.
You must check the interline in the introduction chapter and from lines 318 to 335 and the lenguage.
Reviewer 2 Report
I would recommend this work for publication in Antibiotics after very minor corrections.
Line 30, 32, 36 etc. -- please remove the dots before opening the square bracket . [1]. --> [1].
Line 73-75 -- is that necessary to write about results of this study in the introduction?
Reviewer 3 Report
The manuscript « Antibacterial and anti-inflammatory properties of a novel anti-microbial peptide derived from LL-37”, proposed byZhua and colleagues is reporting the activities of a modified version (KR-12-3) of a short peptide segment (12 amino acids, KR-12) of LL-37 enriched in one basic residue (the acidic residue aspartic acid -D being replaced by the basic amino acid residue arginine -R, and a glutamine residue (Q) replaced by lysine (K) a basic amino acid. The authors made also the choice to replace a basic residue of KR-12 by a large size aromatic amino acid (tryptophan, W) instead of an arginine (R). The novelty of this paper is the design of a new short version of LL-37 and the promotion of relatively positive data on the non-toxicity of the peptide candidate, anti-inflammatory properties and antibacterial activity against Streptococcus gordonii a strain present in the oral cavity.
Comments section per section
Introduction section is properly documented and appropriate to understand the work performed if we except that the authors did not explain the rational of the amino acid replacements they decided. For example why replacing two (Q and D) amino acids by basic residues while at the same time the authors are changing a basic amino acid by a tryptophan, promoting the possibility for the peptide to oxidize and reducing the net charge of their candidate through this replacement. A minor comment line 42 last sentence starting by And needs to be added to the previous sentence. No need also to add the amino acid sequence of KR-12 line 52.
Materials & Methods: In general, this section is starting by a Chemical section. Is there any reason for not providing such important information? Details are not always mentioned in the text. Minor comment the authors need to use the Journal policy for the writing of mL or ml. What is the Mass spectrometry technic for assessing the peptide purity? Did the authors measured the net peptide content to define their initial stock solution or not? After peptide purification the peptide powder is often enriched in TFA salts, this may considerably modify the peptide concentration. The way to write 2 fold serial dilutions from the lowest concentration to the highest one is surprising. Line 129: the authors are mentioning “The concentration of… with the cells was 312.5 ug/ml” It seems to be the MBC? Why not performing the experiment at different concentrations including higher concentrations than the MBC concentration? In table 1, the writing is not homogenous and the referee is recommended to add a column to precise the origin of the markers on the left of the column “Gene”. In this section, it is not clear if the authors are running experiments in duplicates (line 154) or in triplicated (recommended, line 156).
Results
Subsection 3.1 Legent of table 2 should be before the Table itself. In this table KR-12 %of hydrophobic residue is 40% while in the text it is 41.67. What is the correct %? Sentence line 168 “IN the PBS, blabla”. What is the method used to established this? Did the authors performed CD analysis? Or this is just a computer assisted simulation? I tis recommended to add the pI of the molecules in the table. Figure 1 can be removed as I tis not informative. If the editor will recommend to keep it, what is the meaning of the arrow? The N and C termini are too small to be readable.
Regarding 3.2 section, do the authors obtained the same MIC and MBC whatever the replicate? Apparently the selected peptide is really poorly active. DO the authors have a positive control such as any linear peptide known to have activity on Streptococci.
It would have been interesting to extent the activity screening on additional bacterial strains (including clinical isolates and resistant strains) present in the oral cavity, and on yeast cells. Limiting the screening on S. gordonii is fare away to promote this molecule as a potent antibacterial agent. As the authors have access to a synthetic peptide, the quantity available would have been sufficient to extent the activity screening. Regarding the experiment on the activity of KR-12-3 on biofilm, cell count is missing. Line 201, PBS is a buffer and not a solvent. A scale needs to be added in figure 3. Section 3.5 what is the correct p value (text is <0.05, figure P<0.0001)? What is the number of replicates? This information is not included in the figure capture. Figure 6a, the y axis should be pg and not Pg.
Discussion: minor comment, line 284. “some researchers” please specify who are they. Lines 295 and 296 remove “wall” (twise). Line 297. The activity against S. gordoniis” is considered by the authors as excellent. Is really a MBC at >300 ug/mL excellent for developing a drug? The referee is not considering this value as an excellent result. The sections from line 330 to 335 are mentioning pathways that may be involved but there is no data to support this. Next paragraph can be removed as its presence as nonsense in this manuscript.
To conclude, the manuscript needs revision, and is lacking one of the most important issue in such researches, experiments of stability of the peptide in the oral environment where the peptide will have to work. The environment around dental implants is more complicated that the in vitro experiments the authors performed. There is no discussion on this and how the authors will face protease degradation, and the interaction with the microbiote of the oral cavity.
References:
This section needs strong revision as they are too many alterations (e.g. ref 5, 7, 9, 10, 13, 15, 17, 19-22, etc.). No homogeneity, list of authors, etc.
For all these reasons, the paper needs to be strongly improved before publication.
Round 2
Reviewer 3 Report
Point 1. Introduction section is properly documented and appropriate to understand the work performed if we except that the authors did not explain the rational of the amino acid replacements they decided. For example why replacing two (Q and D) amino acids by basic residues while at the same time the authors are changing a basic amino acid by a tryptophan, promoting the possibility for the peptide to oxidize and reducing the net charge of their candidate through this replacement. A minor comment line 42 last sentence starting by And needs to be added to the previous sentence.
Response 1. Thank you for your advice. We have added the last sentence line 42 to the previous sentence.
OK considered
Point 2. No need also to add the amino acid sequence of KR-12 line 52.
Response 2. Thank you for your advice. We have deleted the amino acid sequence of KR-12 line 52.
OK considered
Point 3. In general, this section is starting by a Chemical section. Is there any reason for not providing such important information? Details are not always mentioned in the text.
Response 3. Thank you for your advice. We have ordered the peptides used in our research by the company, so we did not add the chemical details. We revised the sentences this part as “KR-12-3 was synthesized by standard solid phase peptide synthesis using standard fluoro-benzyloxycarbonyl synthesis by Dechi Biosciences (Shanghai, China). The final chimeric peptide (over 95% purity) was purified by high performance liquid chromatography HPLC and characterized by mass spectrometry”.
Not really considered. There is still the chemical section missing. Not only the reagents used ofr the chemical synthesis, this comment was referring to all chemicals used in their studies.
Point 4. Minor comment the authors need to use the Journal policy for the writing of mL or ml.
Response 4. Thank you for your advice. We have unified the writing as mL according to the Journal policy.
OK considered
Point 5. What is the Mass spectrometry technic for assessing the peptide purity?
Response 5. The Mass spectrometry technic for assessing the peptide purity was 95%, and we added in this sentence “The final chimeric peptide (over 95% purity) was purified by high performance liquid chromatography HPLC and characterized by mass spectrometry”.
Not considered. The question was not the purity level but the type of mass spectrometry used for that (ESI, MALDI other?)
Point 6. Did the authors measured the net peptide content to define their initial stock solution or not? After peptide purification the peptide powder is often enriched in TFA salts, this may considerably modify the peptide concentration.
Response 6. Thank you for your advice. We use aliquoted peptide powder for high concentration solution preparation. According to the polypeptide manufacturer's report, after purification, they carry out a series of processes to reduce TFA salts enrichment.
Answer is not appropriate. Apparently the answer to the question of the referee should be No measurement of the NPC. It is not clear what the provider of the peptide did to eliminate the TFA salts. Probably the customer des not know precisely what the provider did.
Point 7. The way to write 2 fold serial dilutions from the lowest concentration to the highest one is surprising.
Response 7. We kept using the 2 fold serial dilutions from the lowest concentration to the highest one in our serial experiments, because this method can keep more peptide. The cost of peptide synthesis was the most part of our research.
The response is not the one expected. In fact the referee is simply requested to rewrite the sentence in order to have the highest concentration appearing first and not he lowest one. To exemplify: 2 fold dilutions from 1000 uM to 125 uM.
Point 8. Line 129: the authors are mentioning “The concentration of… with the cells was 312.5 ug/ml” It seems to be the MBC? Why not performing the experiment at different concentrations including higher concentrations than the MBC concentration?
Response 8. In cytoxicity test we used the MBC as the highest concentration, because we considered the higher concentration than MBC would not use in practical application.
Ok accepted
Point 9. In table 1, the writing is not homogenous and the referee is recommended to add a column to precise the origin of the markers on the left of the column “Gene”.
Response 9. Thank you for your advice. We unified the writing as Palatino Linotype in Table 1. We have add ” Primers for TNF-α, IL-6, IL-10, MMP-9,Arg-1, and iNOS (Table 1) were designed using Oligo Explorer software and synthesized by Sangon Biotech (Shanghai, China)” in the text. Therefore, we did not revise the Table 1.
Ok accepted
Point 10. In this section, it is not clear if the authors are running experiments in duplicates (line 154) or in triplicated (recommended, line 156).
Response 10. Thank you for your advice. The experiments used in this study were all repeated at least 3 times. We deleted the sentence line 154.
Ok revision accepted
Point 11. Subsection 3.1 Legent of table 2 should be before the Table itself. In this table KR-12 %of hydrophobic residue is 40% while in the text it is 41.67. What is the correct %? Sentence line 168 “IN the PBS, blabla”. What is the method used to established this? Did the authors performed CD analysis? Or this is just a computer assisted simulation?
Response 11. Thank you for your advice. We replaced the legend of Table 2 before the Table. And the percentage of KR-12-3 was 41.67%, we corrected the percentage in the Table 2. We also add the pI of the molecules as the reviewer recommended in the Table 2. We did not do the CD analysis and we have deleted the description that “the polypeptide does not have an α-helix structure”.
Ok accepted
Point 12. Figure 1 can be removed as I tis not informative. If the editor will recommend to keep it, what is the meaning of the arrow? The N and C termini are too small to be readable.
Response 12. About Fig 1, we use the program of the HeliQuest to transform amino acid sequences into visual graphics in a way to help us better understand protein structure. Therefore, we considered the figure is meaningful and kept it in the text. We added “The helical wheel projection was performed using online program of the HeliQuest: http://heliquest.ipmc.cnrs.fr . Hydrophobic and hydrophilic amino acids are evenly dis-tributed in the peptide chains” in the text, and revised the fig1 legend as “Sequence and helical wheel diagrams of KR-12 and KR-12-3. The down arrows represent the μH vector. Blue circles refer to hydrophilic residues, pink circles refer to glutamine, red circles refer to Aspartic acid, and yellow circles refer to hydrophobic residues”.
This is the author choose. But this figure is really useless and the editor may decide to keep it or not
Point 13. Regarding 3.2 section, do the authors obtained the same MIC and MBC whatever the replicate? Apparently the selected peptide is really poorly active. DO the authors have a positive control such as any linear peptide known to have activity on Streptococci.
Response 13. Thank you for your advice. According to the MIC and MBC, KR-12-3 showed antibacterial against gordonii. We synthesized dozens of peptides, KR-12-3 showed the better antibacterial and anti-inflammatory activity. So we chose KR-12-3 to further analyze. And we did not choose any positive control, because some peptides have strong antibacterial activity and less anti-inflammatory activity, while other peptides have strong anti-inflammatory activity and less antibacterial activity. Therefore we did not set positive control.
Answer very surprising. The authors should have the expertise to select a positive control. Without that how do they decide if their peptide candidate is “excellent” or better than another one
Point 14. It would have been interesting to extent the activity screening on additional bacterial strains (including clinical isolates and resistant strains) present in the oral cavity, and on yeast cells. Limiting the screening on S. gordonii is fare away to promote this molecule as a potent antibacterial agent. As the authors have access to a synthetic peptide, the quantity available would have been sufficient to extent the activity screening.
Response 14. Thank you for your advice. There are clinical isolates and resistant strains present in the oral cavity, we have only selected one strain at present. On the basis of this sequence, we will further transform and produce short peptides with strong performance. We will conduct experiments on a variety of strains to further improve the antibacterial performance.
This is a generic answer that do not bring any added value to this study
Point 15. Regarding the experiment on the activity of KR-12-3 on biofilm, cell count is missing.
Response 15. Thank you for your advice. We showed the anti-biofilm activity using crystal violet staining after the treated with KR-12-3. We considered the result of the counting was repeated with this part so we did not count the cell.
The referee do not understand the meaning of the author answer
Point 16. Line 201, PBS is a buffer and not a solvent. A scale needs to be added in figure 3. Section 3.5 what is the correct p value (text is <0.05, figure P<0.0001)? What is the number of replicates? This information is not included in the figure capture. Figure 6a, the y axis should be pg and not Pg.
Response 16. Thank you for your advice. We revised the PBS buffer in Line 201 and added a bar in Fig3. And we also added the bar in Fig5. We have revised the p value in section 3.5. The experiments used in this study were all repeated at least 3 times. We added this sentence in figure legends. We revised y axis of Fig 6a.
Ok accepted
Point 17. Discussion: minor comment, line 284. “some researchers” please specify who are they. Lines 295 and 296 remove “wall” (twise).
Response 17. Thank you for your advice. We have revised the sentence as “At the same time, Sorsa et al. have focused on host modulation therapy (HMT).” We have deleted the repeated “cell”.
Ok
Point 18. Line 297. The activity against S. gordoniis” is considered by the authors as excellent. Is really a MBC at >300 ug/mL excellent for developing a drug? The referee is not considering this value as an excellent result. The sections from line 330 to 335 are mentioning pathways that may be involved but there is no data to support this.
Response 18. Thank you for your comments. As mentioned above, we have synthesized many peptides, some with antibacterial activity and some with anti-inflammatory activity. KR-12-3 has both activities and has the best effect on S. gordonii. In clinical application, we hope to obtain a kind of drugs with both bactericidal and anti-inflammatory effects. After root planing, we can carry out local anti-inflammatory and bactericidal effects, so as to improve the healing of deep periodontal pocket. For this purpose, we hope to obtain peptides with dual functions. At present, there are many single functional peptides, especially many single bactericidal peptides. But the single functional peptides cannot solve the clinical practical problems, nor the purpose of our research.
And ??????
Point 19. Next paragraph can be removed as its presence as nonsense in this manuscript.
Response 19. Thank you for your comment. But we did not regard this paragraph was nonsense. As mentioned above, our purpose of this research is to develop a bifunctional peptide which can be used in clinical. Anti-inflammatory property is an important function of bifunctional peptides, so the elaboration of anti-inflammatory activity is also an essential part.
The author decision
Point 20. To conclude, the manuscript needs revision, and is lacking one of the most important issue in such researches, experiments of stability of the peptide in the oral environment where the peptide will have to work. The environment around dental implants is more complicated that the in vitro experiments the authors performed. There is no discussion on this and how the authors will face protease degradation, and the interaction with the microbiote of the oral cavity.
Response 20. Thank you for your advice. We have been keeping trying to decrease the protease degradation for maintaining the performance of peptides. Many drug carriers can reduce the effect of protease degradation on peptides, such as gel, vesicles, nano films, type I collagen membrane. These drug carriers have their own characteristics. A series of experiments need to be carried out to evaluate the combination mode with short peptides with different characteristics. This is the experiment of drug metabolism and another part of important work, so it is not reflected in this paper.
Too bad.
Point 21. This section needs strong revision as they are too many alterations (e.g. ref 5, 7, 9, 10, 13, 15, 17, 19-22, etc.). No homogeneity, list of authors, etc.
Response 21. Thank you for your advice. We revised the references as the journal rules.
Ok, the editorial work will confirm.
Author Response
Please see the attachment.

This manuscript is a resubmission of an earlier submission. The following is a list of the peer review reports and author responses from that submission.
Round 1
Reviewer 1 Report
 This article demonstrated that antibacterial and anti-inflammatory properties of peptide, KR-12-3 which was considered to contribute to success of implant therapy.
 The experimental design and the results seem to be exquisite and valid, however, the following points would be considered to improve this article.
Minor point
- The expression style should be unified or the expression style is not appropriate.
 1) in “Results” section
Ex) Figure 1-a, b, c
“Table” is easier to understand the differences of basic structural features of the two peptides like Table 2.
Ex) Figure 3
a) In the figure legend for Figure 3 (line 199), authors described “As indicated by the arrows,”, however, the arrows are not indicated in photos.
b) There is no explanation for Figure (a) (b) and (c)(d), possibly “0h” for (a)(b) and “4h” for (c)(d), respectively.
2) in “Discussion” section
The sspA/B express on the surface layer of bacterial cells. According to the results of this article, the peptide, KR-12-3 seems to have an influence on bacterial cell wall.
a) Line 284 : Can KR-12-3 disrupt bacterial cell membrane, not cell wall ?
b) Is the word “membrane” that authors used in “Discussion” correct term in almost every place in “Discussion” section ?
Authors also used MC3T3-E1 derived from mouse. It is better to distinguish strictly between “cell membrane” and “cell wall”. Moreover “the biological membrane generally referred (ex. Line279) and “the cell membrane (or wall) as the target of action (ex. Line284) should be also distinguish strictly.
- Dosen’t the peptide KR-12-3 have an influence on the calcification ability of MC3T3-E1cells ?
If authors show the results of this point, reviewer think that the value of this article will further increase. The results of the influence on the calcification ability of MC3T3-E1cells is the point that dentists want to know.
Author Response
We are very grateful to your comments for the manuscript. According to your advice, we amended the relevant part in manuscript. All of your questions were answered one-by-one.
- Thank you for your advice.
1) The information in Figure 1-a, b, c has been replaced by tabular display. Some white arrow signs and explanation for Figure 3 have been added.
2) Our experiments show that peptides alter the morphology of bacterial cell walls. The cell membrane statement is indeed inaccurate. We have corrected the statements in our manuscript. The clerical errors have now been revised.
2. This question is meaningful. Our next plan is to evaluate the effect of peptides on osteoblast calcification and osteoclast activity. The osteogenic properties of the peptides will be comprehensively evaluated.
Reviewer 2 Report
- please rephrase the first paragraph of the introduction
- please add reference in line 42
- last paragraph of the introduction: please mention the purpose of the study earlier in that paragraph
- more elaboration of the possible mechanism of the anti-inflammatory properties of KR- 12-3 (downregulation of IL6, IL8)
- why no exploration of the peptide stability? This could have a big impact on the results.
- the conclusion of the abstract has different meaning from the one in the text
Author Response
Thank you for your advises.
- The first paragraph of the article has been carefully rewritten.
- Necessary reference has been added in line 42.
- Specific purpose of this study has been mentioned in the penultimate paragraph.
- Newly added content provides a detailed review and discussion of possible anti-inflammatory mechanisms of peptides.
- Limited to the experimental conditions, we are unable to detect the stability of the peptide. We are trying to detect the stability of protein in vivo and in vitro. Thank you for your advice.
- The conclusion before and after the article have been unified, and there is no difference now.
Reviewer 3 Report
Good Work. The article is attractive and has an excellent quality.
Author Response
Thank you for your review.